Preservation media, durations and cell concentrations of short-term storage affect key features of human adipose-derived mesenchymal stem cells for therapeutic application

Zhang Fengli 1 2
Ren Huaijuan 1 2
Shao Xiaohu 3
Zhuang Chao 1 2
Chen Yantian ytchen@sjtu.edu.cn 1 2
Qi Nianmin biotech@sjtu.edu.cn 1 2 3
1 Cell Culture and Bioprocess Engineering Lab, School of Pharmacy, Shanghai Jiao Tong University , Shanghai , China
2 Engineering Research Center of Cell & Therapeutic Antibody, Ministry of Education, and School of Pharmacy, Shanghai Jiao Tong University , Shanghai , China
3 China Stem Cell Therapy Co., Limited , Shanghai , China
Flores-Valdez Mario Alberto
Electronic publication date: 2017 May 17
Publication date: 2017
Volume: 5
Electronic Location ID: e3301
Received 2017 Jan 4; Accepted 2017 Apr 11
Copyright: ©2017 Zhang et al.
Copyright year: 2017
Copyright holder: Zhang et al.
License: This is an open access article distributed under the terms of the Creative Commons Attribution License, which permits unrestricted use, distribution, reproduction and adaptation in any medium and for any purpose provided that it is properly attributed. For attribution, the original author(s), title, publication source (PeerJ) and either DOI or URL of the article must be cited.
License URL: https://creativecommons.org/licenses/by/4.0/

Keywords: Adipose-derived mesenchymal stem cells, Preservation media, Durations, Cell concentrations, Short-term storage

Funding: CCBE This work was supported by CCBE. The funders had no role in study design, data collection and analysis, decision to publish, or preparation of the manuscript.

==============================
Background

Adipose-derived mesenchymal stem cells (ADSCs) have shown great potential in the treatment of various diseases. However, the optimum short-term storage condition of ADSCs in 2∼8 °C is rarely reported. This study aimed at optimizing a short-term storage condition to ensure the viability and function of ADSCs before transplantation.

Methods

Preservation media and durations of storage were evaluated by cell viability, apoptosis, adhesion ability and colony-forming unit (CFU) capacity of ADSCs. The abilities of cell proliferation and differentiation were used to optimize cell concentrations. Optimized preservation condition was evaluated by cell surface markers, cell cycle and immunosuppressive capacity.

Results

A total of 5% human serum albumin in multiple electrolytes (ME + HSA) was the optimized medium with high cell viability, low cluster rate, good adhesion ability and high CFU capacity of ADSCs. Duration of storage should be limited to 24 h to ensure the quality of ADSCs before transplantation. A concentration of 5 × 106 cells/ml was the most suitable cell concentration with low late stage apoptosis, rapid proliferation and good osteogenic and adipogenic differentiation ability. This selected condition did not change surface markers, cell cycle, indoleamine 2, 3-dioxygenase 1 (IDO1) gene expression and kynurenine (Kyn) concentration significantly.

Discussion

In this study, ME + HSA was found to be the best medium, most likely due to the supplement of HSA which could protect cells, the physiological pH (7.4) of ME and sodium gluconate ingredient in ME which could provide energy for cells. Duration should be limited to 24 h because of reduced nutrient supply and increased waste and lactic acid accumulation during prolonged storage. To keep cell proliferation and limit lactic acid accumulation, the proper cell concentration is 5× 106 cells/ml. Surface markers, cell cycle and immunosuppressive capacity did not change significantly after storage using the optimized condition, which confirmed our results that this optimized short-term storage condition of MSCs has a great potential for the application of cell therapy.

Introduction

The use of mesenchymal stem cells (MSCs) is a potential regenerative therapeutic strategy because of their regenerative and immune-regulatory properties (Kaplan, Youd & Lodie, 2011). Currently MSCs are widely used in treating various diseases, including immune disorders, degenerative diseases, and tissue injuries (Venkataramana et al., 2010; Wei et al., 2013). Although MSCs can be derived from almost every tissue of the body (da Silva Meirelles, Chagastelles & Nardi, 2006; Kern et al., 2006; Mosna, Sensebé & Krampera, 2010), adipose-derived MSCs (ADSCs) are ideal cells for future use in regenerative medicine due to the high abundance of ADSCs in adipose tissue and the minimal morbidity associated with harvesting MSCs from adipose tissue (Bajek et al., 2016; Gomez-Mauricio et al., 2013).

Large-scale application of MSCs in regenerative medicine demands clinically acceptable “off-the-shelf” cell therapy products. Stem cells cryopreserved using dimethyl sulfoxide (DMSO) are commonly used in regenerative medicine; however, a great number of observed adverse reactions were tenuously or convincingly associated with the cryoprotectant DMSO. Cerebral infarction and myocardial injury occurred in two patients after intravenous injection of autologous stem cells with DMSO (Chen-Plotkin et al., 2007). Neurotoxicity was observed in a patient who suffered from a generalized tonic seizure upon infusion of DMSO-cryopreserved peripheral blood stem cells (Mueller et al., 2007). During the infusion of hematopoietic stem cells without washing the DMSO a patient developed bradycardia, abdominal pain and nausea, and 24 h later he developed anasarca and hypertension (Ruiz-Delgado et al., 2009). Other side effects caused by DMSO include cardiac arrest (Rapoport et al., 1991), severe respiratory arrest (Benekli et al., 2000), paradoxical embolism (Darabi, Brown & Kao, 2005), transient consciousness loss (Schlegel et al., 2009) and so on. There are several alternative cryoprotectants, such as ethylene glycol, methanol and polymer hydroxyethyl starch, but these would cause cell injury and researchers have to focus on how to minimize or eliminate their toxicity (Marquez-Curtis et al., 2015). In addition, sometimes brief (i.e., 24–48 h) storage of MSCs is needed, but cryopreservation of MSCs is not a practical way for brief storage of MSCs (Haack-Sorensen et al., 2007; Lazarus et al., 2005; Kim et al., 2004; Lane et al., 2009). The short-term storage of fresh MSCs in 2∼8 °C does not require cryoprotectants which have underlying safety issues. Also it does not require complicated liquid nitrogen device, which means it can be used to improve the transportability of MSCs products.

In order to maintain high quality of MSCs during the time between harvesting and administration, the surrounding environment needs to be strictly controlled and some key factors must be taken into account (Gálvez-Martín et al., 2014). Several factors including preservation media, durations of storage and cell concentrations may affect the viability and function of MSCs when suspended in liquid storage medium (Lane et al., 2009; Kao, Kim & Daley, 2011; Chen et al., 2013). Different kinds of preservation media including M199 (Mohamadnejad et al., 2007), PBS (Wang et al., 2011), NS (Venkataramana et al., 2010), PlasmalyteA (Chen et al., 2013), 1% HSA in DMEM (Lane et al., 2009), 20% HSA and 5% glucose in Ringer’s lactate (Gálvez-Martín et al., 2014) have been used in previous studies. However, M199 and PBS are not approved vehicles for safe injections thus they could not be used clinically. The viability of cells stored in 1% HSA in DMEM decreased rapidly (Lane et al., 2009). There was no quality evaluation of the cells suspended in NS (Venkataramana et al., 2010). Transplantation of cells immediately after the harvest could receive best clinical outcomes because the quality of cells before administration affects therapeutic efficacy greatly. However, it takes hours or days to progress from harvest to transplantation inevitably (Sohn et al., 2013). Thus, it is of great importance to optimize an appropriate duration of storage with clinically acceptable cell viability and function. Although cell viability in 20% HSA and 5% glucose in Ringer’s lactate was high (>80%) till 48 h, there was no research on the proliferation and immunosuppressive capacity of MSCs (Gálvez-Martín et al., 2014). It has been reported that cell concentrations may affect biological properties of hematopoietic stem cells and cell viability of non-MSC cell lines (De Loecker et al., 1998; Espina et al., 2016). Thus, we also evaluated the effects of cell concentrations during short-term storage on the characteristics of MSCs.

Short-term storage condition with high viability and function of ADSCs has not been studied systematically so far. We aimed to optimize a short-term storage condition to ensure the viability and function of ADSCs for therapeutic application.

Materials & Methods

Study design

This study consisted of four consecutive parts in which preceding results were applied in the subsequent steps. In part I, the impact of different media was measured and the most suitable medium was subsequently used throughout the study. NS and PlasmalyteA are commonly used vehicles, and the supplement of HSA could protect cells from environmental stress and prevent adherence to the tubes or vials (Ikebe & Suzuki, 2014). Dextrose provides a source of energy for cell metabolism (Anderson et al., 1992). Previous study reported that the best preservation medium for short-term storage was 5% dextrose (Pal, Hanwate & Totey, 2008). Thus, we decided to study 5% human serum albumin in 0.9% normal saline (NS + HSA), 5% human serum albumin in multiple electrolytes (ME + HSA, as Baxter Healthcare Co., Ltd. stopped production of PlasmalyteA here in China, ME with totally the same formula as PlasmalyteA was chosen to substitute PlasmalyteA.), dextrose and growth medium (GM) by measuring cell viability and cluster rates, adhesion ability, apoptosis and CFU capacity. In part II, two durations (24 h and 48 h) of storage were evaluated by parameters described above and optimized duration was applied in the following study. In part III, cell concentrations were investigated by adopting measurement of proliferation and differentiation. In part IV, quantification of surface markers, cell cycle, IDO1 gene expression and Kyn concentration of ADSCs suspended in optimized concentration were studied. Cells were stored in 2ml cryogenic vials (Corning Incorporated, Corning, NY, USA) and then placed in a cold chain shipping container designed to ensure stable cooled products transport (2∼8 °C; more than 50 h). A continuous temperature monitoring device was embedded in the cold chain shipping container. ADSCs were suspended after storage in 2∼8 °C for the following research. Unstored cells were fresh cells that did not undergo storage.

Preparation of storage media

5% (500 ml: 25 g) dextrose injection was purchased from Baxter Healthcare (Shanghai) Co., Ltd., China. NS + HSA was prepared by adding 5 ml 20% HSA (Shanghai Institute of Biological Products Co., Ltd., China) into 15 ml NS (Hunan Kelun Pharmaceutical Co., Ltd., China). ME + HSA was prepared by adding 5 ml 20% HSA into 15 ml ME (Sichuan Kelun Pharmaceutical Co., Ltd., China). GM was α-modified minimal essential medium (α-MEM, Gibco, USA) supplemented with 10% fetal bovine serum (FBS, Gemini, Australia).

Isolation and culture of ADSCs

This study was approved by the Ethics Committee of the Shanghai First People’s Hospital at Shanghai Jiao Tong University (number 2013KY080). The 33-year-old woman volunteer signed the contract and permitted adipose tissue to be used for storage and scientific research. The first passage of ADSCs was separated and cultured in GMP condition in Shanghai Kun’ai Biological Technology Co., LTD. Cells were seeded in 6-well plates, cultured with α-MEM supplemented with 10% FBS and 1% penicillin/streptomycin and maintained at 37 °C in a humidified atmosphere at 5% CO2. Medium was changed every three days. Cells were passaged at approximately 80% confluence and passages 3–5 were used for the experiments.

Cell viability and apoptosis

Cell viability and cluster rates were determined on an automatic cell counter (Countstar, Shanghai Ruiyu Biotech Co., Ltd., China) using trypan blue (Gibco, Gaithersburg, MD, USA) staining method. Unstained cells were counted as live cells.

Cell apoptosis was analyzed using a FITC-conjugated Annexin V/PI assay kit (SAB, USA) as our previous study has reported (Ren et al., 2016). 0.5 million cells were rinsed twice with PBS. After centrifugation, 500 µl buffer was added to suspend cells. A total of 5 µl Annexin V-FITC was added to the cell suspension and cells were incubated in the dark for 20 min at 4 °C, followed by addition of 10 µl PI and incubation in the dark for 5 min at 4 °C. Cell apoptosis was determined by flow cytometry (BD Bioscience, San Jose, CA, USA) and analyzed using FlowJo software (TreeStar, Ashland, OR, USA).

Adhesion ability

A total of 1 million cells were seeded in a 100-mm cell culture dish (Corning Incorporated, Corning, NY, USA) and allowed to attach for 24 h at 37 °C in a humidified atmosphere at 5% CO2. Cells were then observed and evaluated under an inverted microscope IX51 (Olympus, Tokyo, Japan).

Colony-forming unit (CFU) capacity

A total of 250 cells suspended in 2 ml α-MEM supplemented with 10% FBS were seeded in a 60-mm dish (Corning Incorporated, Corning, NY, USA) or per well of a six-well plate (Corning Incorporated, USA). After culture for 10 days at 37 °C in a humidified atmosphere at 5% CO2, cells were rinsed twice with PBS and fixed with methanol for 20 min at −20 °C. Then methanol was removed and cells were rinsed twice with PBS. Cells were stained with 1 ml 0.2% crystal violet (Sinopharm Chemical Reagent Ltd., Shanghai, China) for 1 h at room temperature. The plates were rinsed twice with PBS. Stained colonies with >50 cells were scored as CFU and counted under an inverted microscope.

Cell proliferation

Proliferation of ADSCs was assessed by a nontoxic metabolic indicator Alamar Blue (Life Technologies, Carlsbad, CA, USA) as our previous study has reported (Chen et al., 2016). In brief, cells were seeded in a 24-well plate (Corning Incorporated, Corning, NY, USA) at a concentration of 2 × 104 cells/well. After culture for 24 h, culture medium was changed into fresh medium containing 10% (v/v) Alamar Blue indicator and then cells were incubated in the dark for 3 h at 37 °C. Absorbance of the extracted dye was measured by an enzyme immunoassay analyzer (Thermo, USA) at wavelengths of 570 and 590 nm.

Population-doubling time (PDT) was calculated according to the following equation: PDT=t×lg2∕lgNt−lgN0.

In the equation, t indicated duration of proliferation, Nt and N0 represented harvesting cell number and initial seeding cell number, respectively.

Differentiation assay

Adipogenic differentiation

Cells were seeded in a 12-well plate (Corning Incorporated, Corning, NY, USA) at a concentration of 1 × 104 cells/ well and cultured with α-MEM supplemented with 10% FBS until 80% confluency. Cells in differentiation group were incubated with adipogenic induction medium consisted of α-MEM supplemented with 10% FBS, 1 µM dexamethasone (Sigma, St. Louis, MO, USA), 0.5 mM isobutylmethylxanthine (Sigma, St. Louis, MO, USA), 10 µM insulin (Sigma, St. Louis, MO, USA) and 200 µM indomethacin (Sigma, St. Louis, MO, USA). Cells in control group were cultured with α-MEM supplemented with 10% FBS. Medium was changed every three days. After incubation for three weeks, cells were stained by Oil Red O (Sigma, St. Louis, MO, USA) and observed under an inverted microscope. Then, adipogenic differentiation was quantified by an enzyme immunoassay analyzer at 510 nm after the elution with isopropyl alcohol for 10 min at room temperature.

Osteogenic differentiation

Cells were seeded in a 12-well plate at a concentration of 5 × 103 cells /well and cultured with α-MEM supplemented with 10% FBS until 80% confluency. Cells in differentiation group were incubated with osteogenic induction medium consisted of α-MEM supplemented with 10% FBS, 0.1 µM dexamethasone, 10 mM β-glycerophosphate (Sinopharm Chemical Reagent Ltd., Shanghai, China) and 200 µM ascorbic acid (Sigma, St. Louis, MO, USA). Cells in control group were cultured with α-MEM supplemented with 10% FBS. Medium was changed every three days. After incubation for three weeks, cells were stained by Alizarin red (Chroma-Schmidt GmbH, Köngen, Germany) and observed under an inverted microscope. Then osteogenic differentiation was quantified by an enzyme immunoassay analyzer at 570 nm after the solubilization with 10% cetylpyridinium chloride (Sigma, St. Louis, MO, USA) for 10 min at room temperature.

Quantification of surface markers

The surface markers of the cells were examined by flow cytometry. Briefly, 2.5 × 105 cells were rinsed twice with PBS, and then suspended in PBS supplemented with 2% FBS. Cells were stained in the dark for 30 min at 4 °C with the antibodies (BD Biosciences, San Jose, CA, USA): R-phycoerythrin-(PE)-labeled HLA-DR, CD 34, CD45, CD73, CD90 and CD105. The control for PE-coupled antibodies was isotypic mouse IgG1. The data were evaluated using CellQuest software (BD Biosciences, San Jose, CA, USA) and analyzed by FlowJo software.

Cell cycle

The proportion of cells in different phases of cell cycle was analyzed by cell cycle staining buffer (Multisciences Biotech, Hangzhou City, China). Cells were rinsed twice with PBS and then incubated with buffer at a concentration of 1 × 106 cells/ml for 30min in the dark at room temperature. Cell cycle was determined by flow cytometry and analyzed using FlowJo software.

Immunosuppressive capacity

IDO1 is a rate-limiting enzyme in the kynurenine pathway which plays an important role in the induction of immune tolerance. To assess the immunosuppressive capacity of ADSCs, IDO1 gene expression and IDO1 activity assay were performed. Kyn concentration was evaluated as an index which could reflect IDO1 activity.

After storage, cells were seeded in a six-well plate at a concentration of 5 × 105 cells/well. After culture for 24 h at 37 °C in a humidified atmosphere at 5% CO2, culture medium was changed into fresh medium supplemented with 500 U/ml IFN-γ(PeproTech, Rocky Hill, NJ, USA) and then induced for 24 h. In control group, unstored cells were seeded in a six-well plate, and then treated using the same method as stored cells. Cells were used for IDO1 gene expression assay and supernatant was collected for Kyn concentration assay.

IDO1 gene expression

After incubation for 24 h, total RNA was extracted from ADSCs using RNAiso Plus Kit (Takara, Kusatsu, Shiga, Japan) following the manufacturer’s protocol. Total RNA concentrations were quantified using NanoDrop1000 (Thermo, Waltham, MA, USA). 1 µg total RNA was reserved. Real-time polymerase chain reaction (PCR) was achieved by SYBR green system (Takara, Japan). Amplifications for cDNA samples were carried out at 95 °C for 30 s, followed by 40 cycles at 95 °C for 5 s and at 60 °C for 30 s. Primer sequences were listed in Table 1. The relative quantification of target gene was calculated using the 2−▵▵t method and normalized to the transcript levels of glyceraldehyde 3-phosphate dehydrogenase (GAPDH). Melting curve profiles were produced at the end of each PCR to confirm the specific transcriptions of amplification (Chen et al., 2016).

Table 1 Real-time PCR primer sequences.

Gene	Forward primer	Reverse primer	
IDO1	CTGGGCATCCAGCAGACT	TGAGCTGGTGGCATATATCTTCT	
GAPDH	AACAGCGACACCCACTCCTC	CATACCAGGAAATGAGCTTGACAA	

Kyn concentration

A total of 1 ml cell supernatant was mixed with 250 µl 30% trichloroacetic acid (Sinopharm Chemical Reagent Ltd., Shanghai, China) and the mixture was vortexed and centrifuged at 12,000 rpm for 20 min at 4 °C. After filtration with 0.22 µM membrane (Life Sciences, St. Louis, MO, USA), supernatant was analyzed by high performance liquid chromatography LC-20AT (HPLC, Shimadzu, Japan) equipped with a Phenomenex Gemini C18 (250 × 4.6 mm, 5 µm) column. Mobile phase A was 1 µmol/l potassium dihydrogen phosphate (pH = 4; Sinopharm Chemical Reagent Ltd., Shanghai, China) and mobile phase B was methanol (Merck, Kenilworth, NJ, USA). The rate of mobile phase A and B was 3:1 and the flow rate was 1 ml/min. Kyn concentration was detected by the UV-detector at a wavelength of 360 nm at room temperature.

Statistical analysis

All data were shown as mean ± standard deviation, and difference and significance were verified by an one-way analysis of variance (ANOVA), followed by the least-square difference (LSD) for multiple comparisons test. A level of significance of P < 0.05 was used to indicate statistical differences. The statistical analysis was performed using SPSS 19 (SPSS Inc., Chicago, IL, USA).

Results

Part I: evaluation of preservation media

Harvested ADSCs were suspended in dextrose, ME + HSA, NS + HSA and GM at a concentration of 1 × 106 cells/ml. After storage for 24 h at 2∼8 °C, cells were used for the following research. Unstored cells were used as control.

The viability of cells in ME + HSA (95.88 ± 0.69%) and NS + HSA (91.96 ± 1.53%) were significantly higher than in dextrose (67.81 ± 6.37%) (Fig. 1A, P < 0.05). Cell viability in dextrose dropped off dramatically after storage for 24 h. The cluster rate of cells in GM (26.99 ± 1.84%) was significantly higher than that of cells in ME + HSA (3.44 ± 0.81%), NS + HSA (4.23 ± 2.46%) and dextrose (3.82 ± 0.04%) (Fig. 1A, P < 0.05). As shown in Fig. 1A, cells in GM clustered obviously. These results indicated that dextrose, with low cell viability, and GM, with extremely high cluster rate, were not suitable media for the storage of ADSCs.

Figure 1 Optimization of preservation media.

(A) Photos of cells on counting chamber and analysis of viability and cluster rate. White arrow indicated cell cluster. (B) Apoptosis analysis by flow cytometry. (C) Morphology of cells re-plated on 100-mm dish after storage. (D) Photos and analysis of CFU. Results were presented as the means ±  standard deviation for n = 3, ∗P < 0.05.

In our subsequent observation, we found the viability of cells after storage measured by trypan blue staining was not precise and sensitive enough, so we performed Annexin V/PI binding assay (Fig. 1B). The proportions of normal cells in dextrose, ME + HSA, NS + HSA and GM all decreased but there were no significant differences among them. The proportions of early stage apoptotic cells and late stage apoptotic cells both increased in dextrose, ME + HSA, NS + HSA and GM, but there were no significant differences among them, respectively.

The adhesion ability of ADSCs after storage was observed under an inverted microscope (Fig. 1C). Attached cells in the four groups after storage all showed similar spindle-shaped morphologies to cells in the unstored group. However, a mass of detached cells were obviously observed in NS + HSA, which indicated that a lot of cells lost their adhesion ability after storage in NS + HSA.

An evaluation of CFU capacity was performed on ADSCs (Fig. 1D). All groups could form colonies with >50 cells after culture for 10 days. However, the CFU of cells in ME + HSA (13.33 ± 2.05) was significantly higher than that of cells in NS + HSA (2.40 ± 1.06). These results indicated that ME + HSA was a better preservation medium than NS + HSA.

Based on the study of different preservation media, ME + HSA was selected as a proper preservation medium for high cell viability, low cell cluster rate, good adhesion ability and high CFU capacity.

Part II: evaluation of durations of storage

ME + HSA was selected as the storage medium for further study. The storage of ADSCs in ME + HSA for durations of 24 h and 48 h at 4 °C at a concentration of 1 × 106 cells/ml were studied. Unstored cells were used as control.

The cell viability after storage for 48 h (95.34 ± 4.72%) was very high and there was no significant difference compared to cells stored for 24 h (98.11 ± 1.33%), as data were shown in Fig. S1. The cluster rate was lower after storage for 48 h (7.98 ± 1.20%) than after storage for 24 h (15.06 ± 1.34%). It seemed that cells could be stored in ME + HSA with high viability.

Apoptosis was evaluated at 24 h and 48 h after storage (Fig. 2A). The proportion of late stage apoptotic cells increased notably over storage time from 24 h (29.13 ± 3.22%) to 48 h (41.53 ± 1.15%). However, no significant difference in early stage apoptotic cells was shown over storage time from 24 h (19.8 ± 4.16%) to 48 h (21.3 ± 0.36%). These results indicated that extending the duration of storage from 24 h to 48 h would accelerate the apoptosis especially from early to late stage apoptosis.

Figure 2 Optimization of durations.

(A) Apoptosis analysis of ADSCs in different durations by flow cytometry. (B) Morphology of cells re-plated on 100-mm dish. (C) CFU of cells in different durations. Results were presented as the means ± standard deviation for n = 3, ∗P < 0.05.

Although the spindle-shaped morphology of attached cells did not change over the storage time (Fig. 2B), there were significantly fewer attached cells following storage for 48 h than for 24 h. The number of cells lost their adhesion ability increased obviously from 24 h to 48 h.

After storage for 48 h, cells could form colonies with > 50 cells (Fig. 2C); however, the number of these colonies formed after storage for 48 h (8.67 ± 1.67) was obviously lower than that for 24 h (17.07 ± 4.01).

In conclusion, cells could not be stored in ME + HSA for 48 h due to high level of apoptosis, poor adhesion ability and low CFU capacity although viability of cells suspended in ME + HSA for 48 h was very high. 24 h was shown to be an appropriate duration of storage, with relatively low proportion of late stage apoptosis, high adhesion ability and CFU capacity.

Part III: evaluation of cell concentrations

ADSCs suspended in ME + HSA were stored for 24 h at 2∼8 °C at various concentrations: 1 × 106 cells/ml, 5 × 106 cells/ml and 10 × 106 cells/ml. Unstored cells were used as control.

Apoptosis of cells in different cell concentrations was shown in Fig. 3A. The proportion of normal cells decreased obviously as the cell concentration increased from 1 × 106 cells/ml (50.6 ± 3.66%) to 10 × 106 cells/ml (37 ± 0.75%). Cells at a concentration of 5 × 106 cells/ml (30.40 ± 2.87%) showed obviously higher level of early stage apoptosis than cells at 1 × 106 cells/ml (19.80 ± 4.16%) and 10 × 106 cells/ml (23.33 ± 3.66%) (P < 0.05). Cells at concentrations of 1 × 106 cells/ml (29.13 ± 3.22%) and 5 × 106 cells/ml (26.37 ± 7.43%) showed lower level of late stage apoptosis than cells at 10 × 106 cells/ml (40.60 ± 3.78%). These results indicated that cells stored at the concentration of 10 × 106 cells/ml would cause the highest level of late stage apoptosis.

Figure 3 Optimization of cell concentrations.

(A) Apoptosis analysis of ADSCs by flow cytometry. (B) Morphology of cells re-plated on 100-mm dish. (C) CFU of cells at different concentrations. Results were presented as the means ±  standard deviation for n = 3, ∗P < 0.05.

Attached cells of all three groups could form spindle-shaped morphology (Fig. 3B), and it seemed that there were no significant differences in adhesion ability among cells at three cell concentrations.

Cells could form colonies with >50 cells at all three concentrations (Fig. 3C). There were no obvious differences in CFU numbers among cells suspended at 1 × 106 cells/ml (17.07 ± 4.01), 5 × 106 cells/ml (13.00 ± 1.40) and 10 × 106 cells/ml (15.47 ± 1.29). These results indicated that cell concentrations during the storage period did not impact on the CFU capacity.

Proliferation ability was shown in Fig. 4. The fluorescence value reached the peak at the 7th day in unstored cells and the PDT was 57.57 ± 4.77 h. The fluorescence values increased slowly until the 4th day in group 5 × 106 cells/ml and group 10 × 106 cells/ml, and then increased rapidly until reaching the peak on the 9th day with the PDT of 79.05 ± 6.74 h and 81.07 ± 7.84 h, respectively. There was no significant difference between group 5 × 106 cells/ml and group 10 × 106 cells/ml, in terms of either fluorescence values or PDT. As the fluorescence value in group 1 × 106 cells/ml increased slowly with the time, it did not reach its peak before we stopped our measurement on the 10th day. These results indicated that cells in group 1 × 106 cells/ml had low proliferation potential.

All of these three groups at different concentrations showed osteogenic differentiation (Fig. 5A) and adipogenic differentiation abilities (Fig. 5B). Osteogenic differentiation in group 5 × 106 cells/ml (0.09 ± 0.01) was slightly higher than in group 1 × 106 cells/ml (0.07 ± 0.01) or group 10 × 106 cells/ml (0.07 ± 0.01). Adipogenic differentiation ability in group 5 × 106 cells/ml (0.34 ± 0.03) was obviously higher than in group 1 × 106 cells/ml (0.18 ± 0.02) and group 10 × 106 cells/ml (0.16 ± 0.01) (P < 0.05). These results suggested that group 5 × 106 cells/ml had the best differentiation potential.

Figure 4 Proliferation of ADSCs at different cell concentrations.

Results were presented as the means ± standard deviation for n = 3, ∗P < 0.05.

Figure 5 Multidifferentiation of ADSCs at different cell concentrations (A) Osteogenic differentiation. (B) Adipogenic differentiation.

Results were presented as the means ± standard deviation for n = 3,  ∗P < 0.05.

Level of apoptosis increased when the cell concentration increased from 1 × 106 cells/ml to 10 × 106 cells/ml. It seemed there were no obvious differences among the three groups in terms of adhesion ability or CFU capacity. Thus, we adopted the assays of proliferation and differentiation as two further evaluation parameters. Proliferation assay of group 1 × 106 cells/ml showed that low concentration would slower the growth of ADSCs and cells in group 5 × 106 cells/ml showed best osteogenic and adipogenic differentiation potential. These results suggested that 5 × 106 cells/ml was a suitable cell concentration for short-term storage.

Part IV: evaluation of optimized condition

After the selection of preservative media, durations of storage and cell concentrations, we decided to store ADSCs in ME + HSA for 24 h at a concentration of 5 × 106 cells/ml in 2∼8 °C. In order to give a comprehensive evaluation of our optimized condition, we studied the surface markers, cell cycle and immunosuppressive capacity of ADSCs after storage. Unstored cells were used as control.

After storage, HLA-DR, CD34 and CD45 were all negatively expressed (<2%), while CD73, CD90 and CD105 were all positively expressed (>95%, Fig. 6A). These results indicated that the storage of this optimized condition did not affect the expression of surface markers.

Figure 6 Evaluation of optimized solution.

(A) HLA-DR, CD34, CD45, CD73, CD90, CD105 expression. (B) Cell cycle distribution. (C) IDO1 gene expression by RT-PCR. (D) Kyn concentration by HPLC. Results were presented as the means ±  standard deviation for n = 3.

Flow cytometric analysis of cell cycle distribution was shown in Fig. 6B. After the storage, there were no significant differences in G0/G1, S and G2/M compared with unstored cells, respectively. These results indicated that this optimized condition did not change the cell cycle distribution.

There was no significant fold difference in IDO1 gene expression between stored cells (3393.54 ± 653.65) and unstored cells (3654.41 ± 136.30, Fig. 6C). Also, there were no obvious differences in Kyn concentration between unstored cells (25.24 ± 1.75) and stored cells (27.45 ± 2.31, Fig. 6D). These results indicated that this optimized condition did not change gene expression and activity of IDO1.

Discussion

Adipose tissue was considered to be merely a passive energy store in previous years, before ADSCs could be isolated from adipose tissue as a new source of stem cells in 2001 (Zuk et al., 2001). ADSCs are multipotent cells with the ability to differentiate into both mesodermal and non-mesodermal lineages, similar to bone marrow-derived stem cells (BM-MSCs). In addition, ADSCs have a great number of advantages over BM-MSCs. ADSCs could be collected in large quantity with minimal morbidity (Uzbas et al., 2015), and derivation of ADSCs is easier (less invasive) and much more efficient than that of BM-MSCs. Thus ADSCs are attractive stem cells for regenerative application.

In order to manufacture a clinical-scale large number of ADSCs for cell therapy in regenerative medicine and tissue engineering, strictly quality control is required which means that a cGMP-compliant clean room is needed. It seems impossible to build an expensive cGMP-compliant clean room in the hospital set-up, thus cell products should be produced in a central laboratory for up-scaling cells and then transported to the bedside of the patient (Pal, Hanwate & Totey, 2008). Although cryopreservation is an alternative for long-term storage of ADSCs, its requirement for toxic cryoprotectants (i.e., DMSO) and low recovery rate of cells demonstrated that it is not the best or the safest condition for the short-term storage of cell products (Chen et al., 2013; Grein et al., 2010). There is no standard protocol for short-term storage of fresh cells before transplantation as well as no relatively comprehensive evaluation of cell quality after storage. Previous study reported that MSCs stored in saline or dextrose for more than 2 h lost cell viability significantly. MSCs lost CFU capacity and differentiation ability rapidly as storage time increased. Thus duration of storage was limited to 2 h to ensure the quality of MSCs (Sohn et al., 2013). Although Plasmalyte A, 1% HSA and 5% HSA were FDA approved injections and are typically used as preservation media prior to MSCs transplantation, none of these single components supported the survival of MSCs (Chen et al., 2013). No duration information was given in a clinical trial of MSCs stored in saline for the treatment of ischemic stroke (Bang et al., 2005). Cell concentration was considered to have an impact on cell quality, but researchers only measured MSCs counts among cells at the concentrations over a range of 0.5 × 106–20 × 106 cells/ml as the evaluation of cell quality (Lane et al., 2009). Other researchers reported few or no details about preservation conditions or the effect of short-term storage on MSCs (Li et al., 2002; Li et al., 2005; Kim et al., 2006; Shen et al., 2006). The primary objective of this study was to optimize preservation media, durations of storage and cell concentrations of ADSCs to provide a feasible short-term storage condition for cell therapy.

Results showed that cells formed clotted cell pellet after storage in GM for 24 h, which confirmed the natural preference of MSCs to form aggregates (Potapova et al., 2008). Thus it would be unsafe clinically to inject cells suspended in GM. In addition, FBS in GM remains associated with safety issues including transmission of viral disease, anaphylactic reactions and production of anti-FBS antibodies (Ikebe & Suzuki, 2014; Mackensen et al., 2000; Sundin et al., 2007). Therefore, GM is not a suitable preservation medium for the short-term storage of ADSCs. When stored in dextrose, dark blue stained cells could be seen obviously. The viability of ADSCs suspended in dextrose decreased to 67.81 ± 6.37%, which was lower than the minimum viability (70%) acceptable by FDA for cell therapy. The deterioration of ADSCs survival in dextrose may be caused by the low-pH level (3.2–5.5). Additionally, high concentration of 5% dextrose (50 g/L) affects regenerative potential of MSCs and induces replicative senescence (Chen et al., 2013). The adhesion ability and CFU capacity of ADSCs in NS + HSA were obviously lower than in ME + HSA respectively, despite the fact that the viability and apoptosis of cells in NS + HSA had no significant differences compared to cells in ME + HSA respectively. Thus, a comprehensive evaluation system was needed to test the quality of cells before administration as rapid detection of viability and apoptosis would not always be reliable parameters. The mechanism of different effects on cell quality between ME + HSA and NS + HSA was not clear. We supposed that pH of two media may cause the difference as ME has physiological pH (7.4) while NS has lower pH (4.5–7.0). Also the sodium gluconate ingredient in ME could provide energy for cell survival.

Another point to discuss is the proper duration of short-term storage. We thought 24 h was enough for the transport of cell products to another city thousands of kilometers away and for the preparation of both patients and doctors. We also wanted to investigate the longest duration with >70% cell viability. Results showed that the viability of cells in 48 h was high (>90%), however, the late stage apoptosis rate in 48 h was also high (41.53 ± 1.15%). The adhesion ability was poor and CFU capacity was low (8.67 ± 1.67). High level of late stage apoptosis, poor adhesion ability and low CFU capacity showed that cells could not adequately be stored for 48 h. These results may be a consequence of reduced nutrient supply and increase in waste and lactic acid accumulation during prolonged storage (Robinson, Picken & Coopman, 2014). The late stage apoptosis of cells (29.13 ± 3.22%) stored for 24 h was acceptable, and the CFU number (17.07 ± 4.01) of cells stored for 24 h was relatively high. All these results suggested that cells suspended in ME + HSA could be stored for 24 h before administration.

Cell concentration during the storage is also an important factor which may affect cell quality. Compared to intravenous injection, subcutaneous injection and intramuscular injection require higher concentrated cell products with smaller volumes. As previous study has reported (Espina et al., 2016), MSCs suspension injected into small tendinous lesions would leak outside the defect and into the peritendinous tissue even the volume of cell suspension was only 1 ml. Thus cell products of high cell concentration are needed. We compared three cell concentrations 1 × 106 cells/ml, 5 × 106 cells/ml and 10 × 106 cells/ml. Late stage apoptotic rate increased as the cell concentration increased. We didn’t observed significantly differences among these three groups in terms of adhesion ability and CFU capacity. Thus, we adopted the evaluation of proliferation and differentiation. The proliferation of cells stored at a concentration of 5 × 106 cells/ml was very close to that of cells stored at 10 × 106 cells/ml and significantly faster than that of cells stored at 1 × 106 cells/ml. Cells stored at 5 × 106 cells/ml showed the best osteogenic and adipogenic differentiation potential. The high level of late stage apoptosis of cells stored at 10 × 106 cells/ml may be explained that the highest cell concentration may be associated with the fastest lactic acid accumulation (Kilkson, Holme & Murphy, 1984). To some extent, decreasing cell concentration to limit lactic acid accumulation could enhance cell viability and improve cell function (Kao, Kim & Daley, 2011). Long-term proliferation kinetics results indicated that the proliferation potential of cells stored at 1 × 106 cells/ml was impaired. Although increasing cell concentration could improve the proliferation potential, it seemed 5 × 106 cells/ml was high enough as the curve of it was very close to that of 10 × 106 cells/ml. Osteogenic and adipogenic differentiation results also suggested that 5 × 106 cells/ml was a suitable cell concentration. As previous study have reported the same cell concentration of 5 × 106 cells/ml as our result for cell therapy (Garvican et al., 2014; Godwin et al., 2012), we thought cells suspended at 5 × 106 cells/ml could have the highest quality among these three concentrations.

After the evaluation of preservation media, durations of storage and cell concentrations, we thought cells suspended in ME + HSA at a concentration of 5 × 106 cells/ml could be stored for 24 h at 2∼8 °C before administration. We then evaluated this condition by studying surface markers, cell cycle and immunosuppressive capacity of ADSCs after storage. Results showed that surface markers, cell cycle and immunosuppressive capacity did not change after the storage, which confirmed our result that this optimized condition has a great potential for the short-term storage of MSCs for cell therapy.

Conclusions

This is the first study to comprehensively optimize a short-term storage condition of ADSCs. Key factors during short-term storage including preservation media, durations and cell concentrations are studied. Comprehensive evaluation is needed to reflect real cell status before transplantation. Our results show that ADSCs suspended in ME + HSA, at a concentration of 5 × 106 cells/ml could be stored for 24 h at 2∼8 °C, which provides a reliable short-term storage condition for cell therapy. Future studies are still needed to improve cell viability, extend duration of storage, and verify the therapeutic effect of ADSCs after short-term storage in vivo.

Supplemental Information

Figure S1 Photos of ADSCs on counting chamber and analysisof viability and cluster rate in different durations

Results were presented as the means ± standard deviation for n = 3, ∗P < 0.05.

Click here for additional data file.

Data S1 Raw data

Click here for additional data file.

Supplemental Information 1 IDO1 gene

Click here for additional data file.

Supplemental Information 2 GAPDH gene

Click here for additional data file.

We thank Shanghai Ruiyu Biotech Co., Ltd, China for kindly providing an automatic cell counter.

Additional Information and Declarations

Competing Interests

Author Contributions

Human Ethics

DNA Deposition

Data Availability

The authors declare there are no competing interests.

Fengli Zhang conceived and designed the experiments, performed the experiments, analyzed the data, wrote the paper, prepared figures and/or tables, reviewed drafts of the paper.

Huaijuan Ren, Xiaohu Shao and Chao Zhuang contributed reagents/materials/analysis tools, prepared figures and/or tables, reviewed drafts of the paper.

Yantian Chen and Nianmin Qi conceived and designed the experiments, reviewed drafts of the paper.

The following information was supplied relating to ethical approvals (i.e., approving body and any reference numbers):

The Ethics Committee of the Shanghai First People’s Hospital at Shanghai Jiao Tong University (number 2013KY080).

The following information was supplied regarding the deposition of DNA sequences:

Sequences are uploaded as Supplementary Files.

The following information was supplied regarding data availability:

The raw data has been supplied as a Data S1.

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
