# Peer review of "Preservation media, durations and cell concentrations of short-term storage affect key features of human adipose-derived mesenchymal stem cells for therapeutic application"

_PeerJ, doi:10.7717/peerj.3301_

## Round 0.1 · original submission · Major Revisions

Dear author,

After reviewing your work and the commentaries provided by reviewers, particularly 1 and 3, I consider there is an excellent opportunity for you to improve your manuscript by attending these questions and suggestions, therefore I encourage you to resubmit your work after responding to these points.

Reviewer 1 ·

Basic reporting

English writing needs to be improved. There are many errors, some which make sentences and ideas confuse. Context provided but previous work by others is not describe with sufficient detail to clearly detect the contribution of the present study.

Experimental design

The experimental design can be considered simple and logical reasoning for each experiment performed, but not sufficient complete to identify those specific factors that increase or decrease viability and differentiation potential major aspects to make MSC useful for therapeutic applications. This is a technical report, therefore new knowledge is poorly provided. Methods are described in detail but in particular medium called "glucose" is not defined.

Validity of the findings

Conclusions of the study are well established but the meaning of some determinations are not correctly defined. In particular trypan blue staining is used as a measure of viability, because dying cells are not able to exclude the dye. This property is an indication of cell death but not necessarily by apoptosis. Discrepancy between trypan blue staining and Anexin binding, a good indicator of cells dying by apoptosis, is probably due that in some storage conditions cells are dying by mechanism different than apoptosis. It is apparent that some times authors used necrosis to refer to apoptosis. The quality of the work will largely increase if authors identify precisely the cell biological phenomenon affected by storage conditions. This is a superficial study.

Additional comments

Significant improvement in English writing is required. Authors should define the death processes involved such that determinations by trypan blue and anexin binding make sense. "Glucose" medium needs to be defined. A closer comparisons with similar studies, including those based in cryioreservation will increase the relevance of the conclusions.

Reviewer 2 ·

Basic reporting

no comments

Experimental design

no comments

Validity of the findings

no comments

Additional comments

This paper was conducted to optimize types of preservation, time to storage and ADSC concentration to be used in cell therapy. In introduction, It makes a necessary literature review with a solid theoretical foundation about the current context of the subject. It got its purpose and it make emphasizes the importance of this study in the therapeutic application with cellular viability and clinically acceptable functions. Therefore, It represents a relevant contribution and usefull scientific information to be published in this journal.
The structure is adequate and the text has been described with clarity. The purpose were well defined, as well as, a detailed description of the methods used. It makes the possibility to this study to be reproduced as well.
The results are consistent, statistically substantiated, discussed and interpreted in a relevant way, and we believed they can impact on future researches. The illustrations are self-explanatory and they present good quality. The figures and tables relevants. The explanation of figures and references are in accordance with the author guidelines. The conclusions highlight relevant staitments and they are in accordance with the objectives and methods proposed in the research. This paper has important contribution to the scientific comunity.

Reviewer 3 ·

Basic reporting

The main objective of using a preservation medium during freezing process, is because in the clinical application it is logistically complicated to extract, process and use in a single day. For that reason it would consider as objective to find a product that allows the freezing and effective preservation for more than 24 hours.
The structure of the submitted article is acceptable.

Experimental design

The research question is well defined, the methods are described with sufficient detail, and with enough information to be reproducible.

Validity of the findings

The data shown are sufficiently robust, the statistical method employed is correct. There is a degree of novelty and it could be of interest to a specific niche.

Additional comments

In this paper the conclusions are limited to the description of the results, it is opportune to include a deeper analysis of these. The conclusions should be appropriately stated.

---

## Round 0.2 · accepted · Accept

Dear Dr Chen,

I am glad to let you know that your revised version is suitable for acceptance in PeerJ. I appreciate your taking into account the suggestions provided by our reviewers.